# Two Novel SNPs in *RET* Gene Are Associated with Cattle Body Measurement Traits

**DOI:** 10.3390/ani9100836

**Published:** 2019-10-21

**Authors:** Yuan Gao, Bizhi Huang, Fuxia Bai, Fei Wu, Zihui Zhou, Zhenyu Lai, Shipeng Li, Kaixing Qu, Yutang Jia, Chuzhao Lei, Ruihua Dang

**Affiliations:** 1Key Laboratory of Animal Genetics, Breeding and Reproduction of Shaanxi Province, College of Animal Science and Technology, Northwest A&F University, Yangling, Xianyang 712100, Shaanxi, China; gaoyuan710@nwsuaf.edu.cn (Y.G.); 18821700428@163.com (F.B.); 18392360732@163.com (F.W.); 15636110299@163.com (Z.Z.); lzy18408210600@126.com (Z.L.); lsp17782767206@163.com (S.L.); leichuzhao1118@126.com (C.L.); 2Yunnan Academy of Grassland and Animal Science, Kunming 650212, China; hbz@ynbp.cn (B.H.); kaixqu@163.com (K.Q.); 3Institute of Animal Science and Veterinary Medicine, Anhui Academy of Agriculture Science, Hefei 230001, China; yutang2018@163.com

**Keywords:** *RET* gene, SNP, body measurement trait, cattle

## Abstract

**Simple Summary:**

The aim of this study was to identify crucial genes and markers potentially associated with cattle economic traits which would provide a basis for molecular marker-assisted breeding and assist in the genetic selection of cattle. The *RET* gene plays an important role in the development of the gastrointestinal nervous system, which may influence animal body measurement by nutrient absorption. However, there have been no reports on the effects of the *RET* gene on the body measurement traits of cattle. Two novel SNPs (c.1407A>G and c.1425C>G) were identified in this study, which were significantly associated with the body measurement of two Chinese cattle breeds (Qinchuan and Nanyang cattle). The results suggest that c.1407A>G and c.1425C>G can be used in cattle growth-related traits marker-assisted selection breeding.

**Abstract:**

The rearrangement of the transfection (*RET*) gene, which mediates the functions of the ganglion in the gastrointestinal tract, plays an important role in the development of the gastrointestinal nervous system. Therefore, the *RET* gene is a potential factor influencing animal body measurement. The aim of this study was to reveal the significant genetic variations in the bovine *RET* gene and investigate the relationship between genotypes and body measurement in two Chinese cattle breeds (Qinchuan and Nanyang cattle). In this study, two SNPs (c.1407A>G and c.1425C>G) were detected in the exon 7 of *RET* gene by sequencing. For the SNP1 and SNP2, the GG genotype was significantly associated with body height, hip height, and chest circumference in Qinchuan cattle (*p* < 0.05). Individuals with an AG-CC genotype showed the lowest value of all body measurement in both breeds. Our results demonstrate that the polymorphisms in the bovine *RET* gene were significantly associated with body measurement, which could be used as DNA marker on the marker-assisted selection (MAS) and improve the performance of beef cattle.

## 1. Introduction

The intestine is an important organ that maintains nutrient absorption the gut peristalsis reflection mainly regulated by the enteric nervous system (ENS) which is the part of the autonomic nervous system that directly controls the gastrointestinal tract [1,2]. The ENS consist of a collection of autonomic ganglia and associated neural connectives in the wall of the bowel [3]. The ENS interacts with both the gut endocrine and immune systems and has roles in modifying nutrient absorption and maintaining the mucosal barrier [4]. A complete enteric nervous system is necessary for proper gut function; however, there are disorders that arise as a consequence of defective neural crest cell development. This disorder is characterized by sustained contraction of the ganglionic bowel segment, leading to intestinal obstruction and distension of proximal segments (megacolon) [5]. One such disorder is Hirschsprung’s disease (HSCR), also known as congenital megacolon or intestinal aganglionosis, which occurs in one out of 5000 live births, resulting in intestinal obstruction in neonates and megacolon in infants and adults [6]. This disease has been ascribed to the absence of autonomic ganglion cells, which are in the terminal hindgut derived from the neural crest [7]. Moreover, the heritability of HSCR is nearly 100% with clear multigenic inheritance. To date, at least 15 genes have been found to be involved in HSCR, with the major susceptibility gene being the *RET* (the rearrangement of the transfection) [8,9,10]. During development, *RET*-expressing neural crest cells migrate caudally via the intestinal mesenchyme to form ENS, located in the gut wall of the gastrointestinal tract [11].

The *RET* gene encodes a putative receptor tyrosine kinase with a cysteine-rich extracellular domain, a transmembrane domain, and an intracellular tyrosine kinase domain. Germline mutations in the *RET* gene result in the dimerization of spontaneous ligand-independent receptor and autophosphorylation of tyrosine residues [12,13]. The *RET* gene plays a critical role in cell signaling and mutations of the gene lead to disruption of cell proliferation and differentiation of tissues derived from neural crest cells, such as C-cells of enteric autonomous nerve plexus [14,15]. In total, 80% identified mutations of HSCR occurring on *RET* could trigger the defection of intestinal ganglion cell and these mutations located in the coding and/or non-coding regions [9,16,17]. In addition, the mutations of the *RET* have been confirmed to be associated with sporadic and hereditary endocrine tumors [15]. Considering the function of *RET* on gastrointestinal nervous system development, we made the heroic assumption that *RET* may play a crucial role in body measurement by modifying nutrient absorption.

Marker-assisted selection (MAS) has been widely used in breeding. The application of DNA markers for improving growth traits through MAS is a more efficient and powerful tool than the traditional method [18,19]. Therefore, this study aimed to reveal the significant genetic variations in the bovine *RET* gene and investigate the relationship between genotypes and body measurement in two Chinese cattle breeds, Qinchuan cattle (QC) and Nanyang cattle (NY). The results of this study provide a new theoretical basis for cattle breeding and genetics. 

## 2. Materials and Methods

### 2.1. Ethics Statement

The experimental animals and procedures performed in this study were approved by the Faculty Animal Policy and Welfare Committee of Northwest A&F University under contract (NWAFU-314020038). The care and use of experimental animals fully complied with local animal welfare laws, guidelines and policies.

### 2.2. Animals and Samples

A total of 342 Chinese indigenous cattle were used in this study, including Qinchuan cattle (QC, *N* = 225, Shaanxi Province) and Nanyang cattle (NY, *N* = 117, Henan Province). The hair samples of 225 female Qinchuan cattle under the same conditions were selected from the Yangling Qinchuan cattle breeding center. The blood samples of 117 female Nanyang cattle were collected from the breeding center of Nanyang cattle where animals were raised under the same forage and feeding management conditions. All the individuals selected were between 3 and 4 years old, healthy, and unrelated. The data of body measurement were recorded for an association analysis, including body height (BH), hip height (HH), body length (BL), chest circumference (ChC), abdomen circumference (AC), hip width (HW), hucklebone width (HuW), rump length (RL), and body weight (BW). These traits were measured using the methods described by Gilbert et al. [20]. 

### 2.3. DNA Isolation and Genomic DNA Pools Construction

Genomic DNA was isolated from a hair sample. To extract DNA from hair, five to eight hair follicles were washed by 75% alcohol and ddH_2_O, digested in 0.5 µl proteinase K (20mg/mL) and 100 µL hair buffer stock (containing 250 µL Tween20, 5 mL 10× PCR Buffer, 5 mL MgCl_2_, 20 µL CaCl_2_, and ddH_2_O) of 65 °C for 40 min and 94 °C for 10 min. Genomic DNA from the blood samples was isolated from leukocytes using a whole-blood genomic DNA extraction kit (Aidlab Biotechnologies Co., Ltd., China). DNA quantity and purity (OD_260_ = OD_280_) for each sample was assessed by a NanoDrop™ 1000 spectrometer (Thermo Scientific, Waltham, MA, USA). DNA samples were diluted to a standard concentration (50 ng/µL) and stored at −80 °C. To explore the allele variation of the bovine *RET* gene, all 30 samples were mixed and used for polymerase chain reaction (PCR).

### 2.4. SNP Detection and Genotyping

To investigate the genetic variation, 18 PCR primer pairs (Table 1) were designed by Primer Premier Version 5.0 (Premier Biosoft International, USA) to amplify the coding sequences of *RET* (GenBank accession number: AC_000185.1). The PCR amplification was performed with 25 μL of reaction volume containing 1 µL genomic DNA (50 ng/µL), 1 µL of each primer (10 µmoL/µL), 12.5 µL Taq Master Mix (Beijing ComWin Biotech Co., Ltd., China) and 9.5 µL ddH_2_O. The amplification conditions of PCR were listed as follows: denaturing at 95 °C for 5 min, followed by 35 cycles of 95 °C for 40 s, annealing at annealing temperature (Table 1) for 40 s, and extending at 72 °C for 30 s, followed by a final extension at 72 °C for 10 min. The PCR products were electrophoresed on 1% agarose gel. PCR products were sequenced directly by Sangon Biotech (Shanghai) Company (China) to identify potential mutation sites.

### 2.5. Statistical Analysis

The sequencing results were analyzed with DNASTAR 5.0. Allele and genotype frequencies were estimated by direct counting. The Chi-squared (χ^2^) test for the Hardy-Weinberg equilibrium (HWE) was applied to assess the deviations of the number of observed versus expected genotypes. Population genetic indexes, such as gene heterozygosity (He), effective allele numbers (Ne) and the polymorphism information content (PIC) were analyzed according to Nei’s methods [21]. The linkage disequilibrium (LD) structure was performed using SHEsis software (http://analysis.bio-x.cn/myAnalysis.php) [22]. For missense mutations, the secondary mutant and normal structures were analyzed using the SOPMA program online (https://npsa-prabi.ibcp.fr/cgi-bin/npsa_automat.pl?page=npsa_sopma.html). The 3D structures were predicted with the SWISS-MODEL (http://swissmodel.expasy.org/). For synonymous mutations, the codon usage frequencies of the target gene were calculated using the Codon Usage Database (www.kazusa.or.jp/codon/countcodon.html).

All the cattle used in this study were unrelated, 3–4 years old, healthy, non-pregnant females, and the different breeds were raised in their respective farms and analyzed separately. Therefore, this study used the reduced linear model to determine the relationship between genotypes and the various body measurement traits. The structure of the model is *Y_i_* = *µ* + *G_i_* + *e*, where *Y_i_* is the phenotypic observations, *µ* is the mean of the phenotypic observation, *G_i_* is the effect of genotype, and *e* is the residual effect [23]. SPSS software (version 18.0) (International Business Machines, Armonk, NY, US) was used to calculate the associations analysis by one-way ANOVA followed by post hoc multiple comparisons and residuals’ normality assumptions were tested using the graphical methods by a histogram and a normal P-P plot of regression standardized residual. The results are presented as means ± SE.

## 3. Results

### 3.1. Identification of Genetic Variants in RET

In this study, two SNPs were identified in exon 7, (SNP1: rs110630023, c.1407A>G) and (SNP2: rs109861339, c.1425C>G)(Figure 1). SNP1 is a missense mutation and resulted in a substitution of Met to Ile at codon 469. Prediction of the secondary structure with SOPMA program indicated that its normal type contained 30.5% alpha helix, 23.79% extended strand, 8.86% beta turn and 36.85% random coil, whereas the mutation found in our study was composed of 30.6% alpha helix, 23.6% extended strand, 8.68% beta turn and 37.13% random coil. The predicted protein 3-D structure of normal and mutant *RET* is shown in Figure 2. The predicted 3D structure remained unchanged (Figure 2a,b). However, there was an obvious difference in the value of quality model energy analysis (QMEAN) between wild type and mutant type (Figure 2e,f). The conversion of Met at the 469 sites to Ile led to the rise of the value of qualitative model energy analysis (QMEAN) from −5.42 to −5.70. Additionally, the solvation and the torsion of the wild type was lower than those of the mutant type, and the Cβ and all atoms of wild type were higher than those of the mutant type, which indicates that the protein of wild type might have a lower residue embedding ability and torsion angle as well as a stronger C-β potential energy of interaction and paired atom distance dependent potentials than the mutant type. SNP2 is a synonymous mutation which caused the 475th codon (Pro) changes from CCC to CCG. The codon usage frequencies of the *RET* gene are shown in Appendix A. The CCG and CCC were the high-frequency and low-frequency codon of Pro, respectively, indicating a possible change of the expression levels of *RET* affected by SNP2.

### 3.2. Genetic Characteristics of SNPs Detected in Bovine RET Gene

For the two detected SNPs, 342 individuals were genotyped by sequencing. The genotyping results are shown in Figure 1. In the two cattle breeds, genetic diversity of bovine *RET* was evaluated using genotypic and allelic frequencies and three genetic indexes (He, Ne and PIC) of two SNPs (Table 2). For SNP1, the AA genotype was dominant, whereas the GG genotype was rare. Heterozygosity (He) was 0.3911 and 0.2484 and the effective allele numbers (Ne) were 1.6423 and 1.6423. The A allele (0.7333–0.8547) was higher than the D allele (0.2667–0.1453) in the two breeds. In SNP2, the C allele was most prevalent (0.7378–0.8974) and the CC genotype was more frequent than the other genotypes. The values of He were 0.3869 and 0.1841, and the values of Ne were 1.6311 and 1.2256 in the two breeds. In both loci, the NY breed belonged to a low polymorphic locus (0 < *PIC* < 0.25) and the QC breed belonged to a moderate polymorphic locus (0.25 < *PIC* < 0.50). They were both in the Hardy–Weinberg equilibrium (HWE) (*p* > 0.05).

To reveal the linkage relationships among SNP1 and SNP2, linkage disequilibrium between these two SNPs was estimated in the QC and NY breeds (Appendix A). In QC cattle, the D’ value was 1.000, and the r^2^ value was 0.977, and in NY, the D’ value was 0.949, and the r^2^ value was 0.605. The results suggest that SNP1 and SNP2 were in a relatively strong linkage in the QC and NY breeds. The haplotype analysis showed that three different haplotypes were identified among the two SNPs (frequencies < 0.03 had been ignored in the analysis). In the QC breed, Hap1 had a higher haplotype frequency (0.733) than Hap2 (0.262). In the NY breed, Hap1 had the highest haplotype frequency (0.850), followed by Hap 2 (0.098) and Hap 3 (0.047) (Table 3).

### 3.3. Association Analysis between Polymorphisms and Body Measurement Traits

The results of the association analysis of the single polymorphic locus with body measurement traits listed in Table 4 and Table 5 reveal that the loci of SNP1 and SNP2 had a significant association with body measurement. The residual normality assumption was tested in Appendix A, which show that one-way ANOVA could be applied in the association analysis. For the SNP1 locus, the body height and hip height of the GG genotype were significantly longer than the AA genotype in the QC population. Moreover, significant differences in chest circumference (*p* < 0.05) were observed between individuals with genotypes AA, CG, and GG in the QC population. In the NY population, individuals with the AA genotype had a significantly longer chest circumference and hucklebone width (*p* < 0.05) than those with AG genotypes. For the SNP2 locus, individuals with the GG genotype showed a significant advantage in body height and hip height compared with those with the AA genotype in the QC population (*p* < 0.05). In addition, the GG and CG genotypes showed a significantly higher chest circumference than CC (*p* < 0.01). In NY cattle, the abdomen circumference of the GG genotype was significantly longer than the CC genotype.

Considering the strong linkage between SNP1 and SNP2, the relationship between the combined genotypes of the SNP1 and SNP2 and the body measurements of the QC and NY cattle were analyzed (Table 6 and Table 7). In the QC population, individuals carrying the GG-GG genotype had a significantly greater body height than those with genotypes AA-CC and AG-CC (*p* < 0.05) and the individuals with the GG-GG genotype had significantly greater hip height than genotype AG-CC (*p* < 0.05). The chest circumference of cattle with the GG-GG and AG-CG genotypes was significantly greater than those with the AA-CC and AG-CC genotypes. In the NY cattle, body height, hip width, hucklebone width, and hip height of the AA-CC and AG-CG genotype were significantly longer than the AG-CC genotype (*P* < 0.01 or *p* < 0.05). Individuals with the AG-CC genotype were significantly shorter than those with the AA-CC, AG-CG, and GG-GG genotype in abdomen circumference (*p* < 0.01). In addition, individuals carrying the AA-CC genotype had significantly greater chest circumference than those with the AG-CC genotype (*p* < 0.05).

## 4. Discussion

Candidate gene is a useful method to investigate the association between phenotypes and genotypes, which is helpful for improving important economical traits in beef cattle [24,25,26]. Recent data have shown that effective molecular markers used in the selection of beef quality traits could bring benefits in the beef industry [27].

In previous studies, the *RET* gene was considered as a major disease-causing locus in HSCR [28]. Some studies have shown that the SNPs of *RET* were relevant to HSCR [29,30,31]. c135G>A and c2307T>G were found in Spanish Hirschsprung patients and had significant correlations with the pathogenesis of HSCR [29]. The result of the c135G/A polymorphism is consistent with the findings in patients from Germany [30]. Sancandi et al. [31] found that two novel SNPs were located at −1 base pair (bp) and −5 bp from the RET transcription start site (−5G>A and −1C>A), the allelic frequencies of which were significantly different between the patients with Hirschsprung’s disease and the controls. In addition, the SNPs located in the promoter and intron of the *RET* were significantly associated with HSCR by influencing the expression of *RET* [17]. However, there was no study on marker-assisted selection in bovine *RET* gene. Therefore, we further verified the relationship between two SNP loci in bovine *RET* gene and their body measurements.

In this study, the genetic diversity of the bovine *RET* gene and the significant association with bovine body measurement traits were researched for the first time. Two SNPs were detected in 342 Chinese cattle. SNP1 (rs110630023: c.1407A>G) is a missense mutation. The normal and mutant *RET* protein secondary structure showed that the alpha helix, extended strand, beta turn, and random coil changed little. The perdition of the three-dimensional structure showed that 3-D structures were unchanged, but there was an obvious difference in the value of the quality model energy analysis (QMEAN) between wild-type and mutant-type. SNP2 (rs109861339: c.1425C>G) is a synonymous mutation. The mutant CCG codon was used more frequently for Pro. For the SNP1, individuals with the GG genotype showed an advantage for body measurement traits in Qinchuan cattle, especially body height, hip height, and chest circumference, compared with those with AA and AG genotypes. In the NY population, individuals with the AA genotype had a significantly longer chest circumference and hucklebone width than those with the AG genotypes; however, the trait gap between individuals with the AA genotype and the GG genotype was small. The χ^2^ test showed that this SNP was in the Hardy-Weinberg equilibrium in different cattle breeds. Therefore, these loci did not undergo external selection. The GG genotype is a rare genotype among breeds with SNP1 locus and can be selected for increasing growth properties. The loci of SNP2 had a significant association with body height, hip height and chest circumference in QC cattle (*p* < 0.05), which were similar to the result of SNP1. In the NY cattle, the abdomen circumference of the GG genotype was significantly longer than the CC genotype.

Based on two popular measures (r^2^ and D’) performed in the linkage disequilibrium analysis, SNP1 and SNP2 in the QC and NY populations demonstrated a strong linkage. Therefore, associations of combined sites with body measurement traits were analyzed. In the QC population, there was a significant difference between the combined genotypes and body height, hip height and chest circumference, which indicated that the cattle with genotype GG-GG could be selected to obtain higher body measurement traits. This result is consistent with the analysis of the single site. In the NY population, significant differences were found in body height, chest circumference, abdomen circumference, hip width, hucklebone width, and hip height (*p* < 0.01 or *p* < 0.05). Although the trait value of the GG-GG genotype was similar to the dominant genotype, there was no significant difference found in body height, hip width, hucklebone width, and hip height. This could be because of the low number (2) of individuals with the GG-GG genotype. Interestingly, a surprising finding was that the individuals with the AG-CC genotype had the lowest values of all body measurement traits in both breeds.

In general, the SNP1 and SNP2 within *RET* gene was significant associated with body measurements, meaning that these two SNP loci could be used as DNA markers for selecting excellent individuals in marker-assisted selection (MAS) breeding, and the *RET* gene could be used as a candidate gene for breeding beef cattle.

## 5. Conclusions

This research investigated SNP in the bovine *RET* gene and two SNPs were detected in exon 7. Our results show that the polymorphism of c.1407A>G and c.1425C>G of *RET* were significantly associated with the body measurements of cattle (*p* < 0.05). Therefore, the bovine *RET* gene could be used as a DNA marker related to body measurement for marker-assisted selection (MAS) and to improve the performance of beef cattle.

## Figures and Tables

**Figure 1 animals-09-00836-f001:**
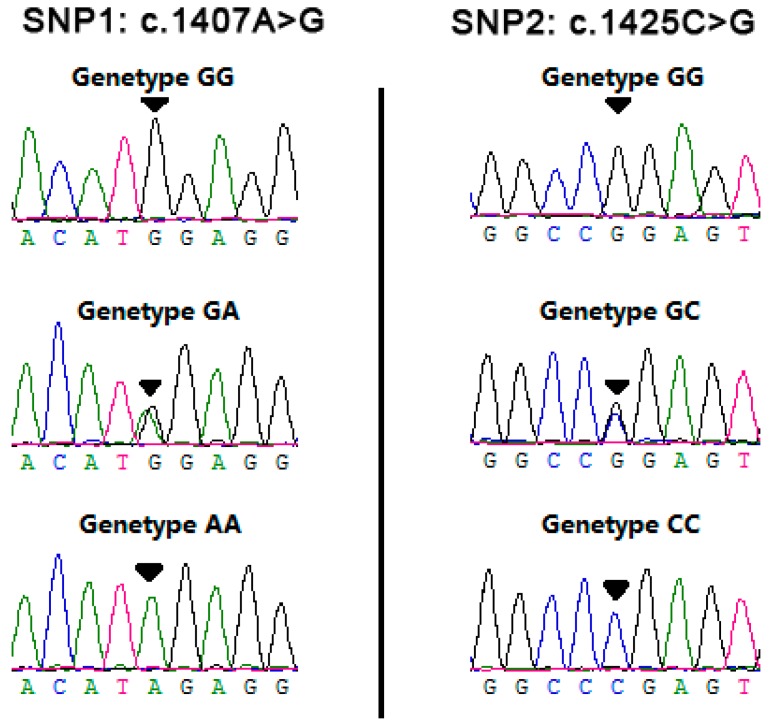
The sequencing results of the bovine *RET* gene and two SNPs are indicated by arrows.

**Figure 2 animals-09-00836-f002:**
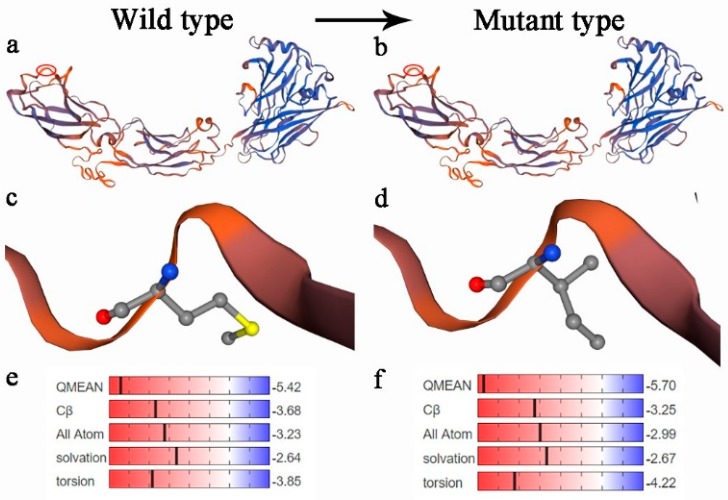
The 3D structure of the bovine *RET* gene was constructed using the SWISS-MODEL. Note: The predicted 3D structure of the bovine *RET* gene of the wild type (**a**) and the mutant type (**b**). The structure of Met (**c**) and Ile (**d**). the qualitative model energy analysis of the wild type (**e**) and the mutant type (**f**).

**Table 1 animals-09-00836-t001:** Primer information of the bovine *RET* gene.

Primer Name	Primer Sequences (5’–3’)	Tm (°C)	Length (bp)	Location
R1	F:CGGCTCTGGTCTCCTAAC R: GCCACACAACCCCACT	51	609	Exon1
R2	F:GCTGGGAAAGCTGATCTGC R:AAGGAGGGACAAGGAGAGG	56	440	Exon2
R3	F:TGTGAGGACAAGGGAGCA R:AATGGATGATGGGGTGGT	63	633	Exon3
R4	F:CTGTGCTTGGAGGCTGTG R:ATGGAGGGACCCAGATGC	63	477	Exon4
R5	F: CCTATGGGCTCTGGCTTTC R:AAGGCTGGGAAGAGGGTT	63	396	Exon5
R6	F:ATGTGTCCAGGGAATGCT R: CTGGCACTGCTCTTTCAT	57	596	Exon6
R7	F:CTGTCCTGGGACTCAAGCTG R:TACAGGACCGCACCCTTCTA	64	438	Exon7
R8	F: CGCAAAGCAGGTATTCAC R: GAGCACAGAGGAAAACT	51	536	Exon8
R9	F: AGCCTCGTTTGGTCTCC R: GAATGTGGGTCAAGCCG	61	578	Exon9
R10	F: GAGTGGGCTGCTGAGTG R: AGGGAGGCATGAGGATC	56	212	Exon10
R11	F: TGGCTCTGTTGGGAGTTGG R: TGAAAGGCAGCACGGTG	56	523	Exon11
R12	F: TGAGCATACGGAGTCCAGC R: CACACCAGCCCACATCAT	58	591	Exon12
R13	F: ATAGCCCAGCCCAGAAGG R:AGTAGGTGCCCAGGAGACA	60	699	Exon13
R14-15	F: AGGAACCTGGCGGGCTAT R: GGGGATGTGGACTGGAAG	56	762	Exon14- Exon 15
R16	F: TTGGGAAGTGGTGCGTG R: GGTCATTCTGGACGGTTGG	58	454	Exon16
R17	F: ACCGTGGGCACTTGGAC R: CCCCAGCATCTGCCATC	62	465	Exon17
R18	F:GCTTGGCAGGTGTTTGTG R:GAAGGATAAAGGCTCGTGC	58	542	Exon18
R19	F: GCGAGAGGTTGATAGGTGC R: CGCCCTCTCCTCTAACATCT	56	544	Exon19

**Table 2 animals-09-00836-t002:** Genotyping and population genetic analysis in *RET*.

Name	Breeds (Sizes)	Genotypic Frequencies			Allelic Frequencies		Ne	He	PIC	χ^2^ (HWE)
		AA	AG	GG	A	G				
c.1407A>G	QC/225	0.5200	0.4267	0.0533	0.7333	0.2667	1.6423	0.3911	0.3146	1.8595
	NY/117	0.7265	0.2564	0.0171	0.8547	0.1453	1.3304	0.2484	0.2175	0.1225
		CC	CG	GG	C	G				
c.1425C>G	QC/225	0.5289	0.4178	0.0533	0.7378	0.2622	1.6311	0.3869	0.3121	1.4308
	NY/117	0.8205	0.1538	0.0256	0.8974	0.1026	1.2256	0.1841	0.1671	3.1578

Note: HWE, Hardy–Weinberg equilibrium; PIC, polymorphism information content; He, heterozygosity; Ne, effective allele numbers; df = 2, χ^2^ 0.05(2) = 5.99, χ^2^ 0.01(2) = 9.21, χ^2^ > 5.99 means deviating from HWE with significant level. χ^2^ > 9.21 means deviating from HWE at a highly significant level; low polymorphism if PIC value < 0.25, moderate polymorphism if 0.25 < PIC value < 0.50, and high polymorphism if PIC value > 0.50.

**Table 3 animals-09-00836-t003:** Haplotypes of the two SNPs in bovine *RET* gene.

Breed	Haplotype	SNP1	SNP2	Frequency
QC	Hap1	A	C	0.733
	Hap2	G	G	0.262
NY	Hap1	A	C	0.850
	Hap2	G	G	0.098
	Hap3	G	C	0.047

Note: frequency < 0.03 was ignored in the analysis.

**Table 4 animals-09-00836-t004:** Association analysis of single SNP in *RET* with body measurements in Qinchuan cattle.

Genotypes	BH (cm)	HH (cm)	BL (cm)	ChC (cm)	RL (cm)
AA	129.92 ± 0.50 ^a^	129.30 ± 0.48 ^a^	154.18 ± 0.90	198.52 ± 1.27 ^a^	50.86 ± 0.31
AG	130.51 ± 0.48 ^ab^	129.64 ± 0.50 ^ab^	156.57 ± 0.91	202.67 ± 1.17 ^b^	50.50 ± 0.37
GG	133.67 ± 2.12 ^b^	132.89 ± 2.33 ^b^	159.44 ± 4.95	208.33 ± 2.55 ^b^	51.33 ± 0.97
*p* vaule	0.035	0.042	0.109	0.011	0.645
CC	129.72 ± 0.50 ^a^	129.18 ± 0.47 ^a^	153.67 ± 0.95	197.80 ± 1.34 ^a^	50.84 ± 0.31
CG	130.66 ± 0.49 ^ab^	129.71 ± 0.52 ^ab^	156.78 ± 0.85	203.09 ± 1.16 ^b^	50.53 ± 0.38
GG	133.67 ± 2.12 ^b^	132.88 ± 2.33 ^b^	159.44 ± 4.95	208.33 ± 2.55 ^b^	51.33 ± 0.97
*p* vaule	0.027	0.036	0.091	0.003	0.622

Note: Values are shown as the least-squares mean ± standard error. ^a,b^ means with different superscripts within the same line represented significantly different (*p* < 0.05). The *p* values are the results of the one-way ANOVA analysis. BH = body height, HH = hip height, BL = body length, Chc = chest circumference, RL = rump length.

**Table 5 animals-09-00836-t005:** Association analysis of single SNP in *RET* with body measurement in Nanyang cattle.

Genotypes	BH (cm)	BL (cm)	ChC (cm)	AC (cm)	HW (cm)	HuW (cm)	HH (cm)	BW (cm)
AA	128.95 ± 0.97	140.79 ± 1.30	176.06 ± 1.37 ^a^	221.95 ± 4.15	46.92 ± 0.76	27.67 ± 0.35 ^a^	131.27 ± 1.78	410.45 ± 11.33
AG	126.83 ± 1.92	137.33 ± 2.41	169.48 ± 3.07 ^b^	207.95 ± 6.57	43.84 ± 1.72	26.00 ± 0.63 ^b^	125.50 ± 3.29	382.61 ± 17.62
GG	127.50 ± 1.50	140.00 ± 9.00	175.50 ± 0.50 ^ab^	234.50 ± 1.50	46.00 ± 1.00	28.00 ± 0.41 ^ab^	130.00 ± 1.00	416.00 ± 11.00
*p* vaule	0.334	0.267	0.050	0.138	0.174	0.043	0.157	0.227
CC	127.97 ± 0.99	139.92 ± 1.30	174.85 ± 1.42	216.21 ± 4.29 ^a^	45.56 ± 0.90	27.37 ± 0.36	128.90 ± 1.86	403.81 ± 10.90
CG	123.88 ± 2.34	139.80 ± 3.24	172.41 ± 4.33	222.22 ± 4.96 ^ab^	47.50 ± 1.36	26.83 ± 0.66	131.55 ± 3.47	402.28 ± 25.32
GG	128.33 ± 1.20	139.83 ± 5.19	176.83 ± 1.36	234.50 ± 1.50 ^b^	46.00 ± 1.00	28.33 ± 0.33	130.00 ± 1.00	410.33 ± 8.51
*p* vaule	0.743	0.974	0.778	0.001	0.626	0.722	0.816	0.992

Note: Values are shown as the least-squares mean ± standard error. ^a,b^ means with different superscripts within the same line represented significantly different (*p* < 0.05). The *p* values are the results of one-way ANOVA analysis. BH = body height, BL = body length, Chc = chest circumference, AC = abdomen circumference, HW = hip width, HuW = hucklebone width, HH = hip height, and BW = body weight.

**Table 6 animals-09-00836-t006:** Association analysis of combined genotypes in *RET* with body measurement traits in Qinchuan cattle.

Genotypes (Number)	BH (cm)	HH (cm)	BL (cm)	ChC (cm)	RL (cm)
AA-CC (113)	129.88 ± 0.51 ^a^	129.36 ± 0.48 ^ab^	154.14 ± 0.93	198.44 ± 1.31 ^a^	50.87 ± 0.31
AA-CG (4)	131.00 ± 2.38 ^ab^	127.50 ± 3.61 ^ab^	155.25 ± 1.03	200.75 ± 3.85 ^ab^	50.75 ± 1.37
AG-CC (6)	126.66 ± 2.10 ^a^	125.83 ± 1.55 ^a^	146.00 ± 7.15	185.83 ± 8.73 ^a^	50.20 ± 1.31
AG-CG (90)	130.65 ± 0.51 ^ab^	129.82 ± 0.51 ^ab^	156.86 ± 0.89	203.21 ± 1.20 ^b^	50.51 ± 0.39
GG-GG (12)	133.67 ± 2.12 ^b^	132.89 ± 2.33 ^b^	159.44 ± 4.95	208.33 ± 2.55 ^b^	51.33 ± 0.97
*p* vaule	0.044	0.033	0.143	0.002	0.922

Note: Values are shown as the least-squares mean ± standard error. ^a,b^ means with different superscripts within the same line represented significantly different (*p* < 0.05). The *p* values are the results of the one-way ANOVA analysis. BH = body height, HH = hip height, BL = body length, Chc = chest circumference, RL = rump length.

**Table 7 animals-09-00836-t007:** Association analysis of combined genotypes in *RET* with body measurement in Nanyang cattle.

Genotypes (Number)	BH (cm)	BL (cm)	ChC (cm)	AC (cm)	HW (cm)	HuW(cm)	HH (cm)	BW (cm)
AA-CC (85)	128.95 ± 0.97 ^a^	140.79 ± 1.30	176.06 ± 1.37 ^a^	221.95 ± 4.15 ^a^	46.92 ± 0.76 ^a^	27.67 ± 0.35 ^a^	131.27 ± 1.78 ^a^	410.45 ± 11.33
AG-CC (11)	120.15 ± 3.99 ^b^	132.50 ± 5.05	164.55 ± 6.24 ^b^	194.12 ± 10.10 ^b^	39.14 ± 2.45 ^b^	24.25 ± 1.35 ^b^	119.62 ± 4.59 ^b^	345.23 ± 34.57
AG-CG (18)	129.88 ± 2.34 ^a^	139.80 ± 3.24	172.41 ± 4.33 ^ab^	222.22 ± 4.96 ^a^	47.50 ± 1.36 ^a^	26.83 ± 0.66 ^a^	131.55 ± 3.47 ^a^	402.28 ± 25.17
GG-GG (2)	127.50 ± 1.50 ^ab^	140.00 ± 9.00	175.50 ± 0.50 ^ab^	234.50 ± 1.50 ^a^	46.00 ± 1.00 ^ab^	28.00 ± 0.01 ^ab^	130.00 ± 1.00 ^ab^	416.00 ± 11.00
*p* vaule	0.043	0.282	0.018	0.008	0.004	0.013	0.025	0.359

Note: Values are shown as the least-squares mean ± standard error. ^a,b^ means with different superscripts within the same line represented significantly different (*p* < 0.05). The *p* values are the results of the one-way ANOVA analysis. BH = body height, BL = body length, Chc = chest circumference, AC = abdomen circumference, HW = hip width, HuW = hucklebone width, HH = hip height, and BW = body weight.

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
