# Peer review of "Two Novel SNPs in RET Gene Are Associated with Cattle Body Measurement Traits"

_animals, 2019, doi:10.3390/ani9100836_

Round 1

Reviewer 1 Report

Review comment

Manuscript ID : animals-606961

Title : Two novel SNPs in RET gene are associated with cattle body measurement traits

General comments

This study reports an association of two SNPs in RET gene with body measurement in Chinese cattle. The SNPs were novel and they had an association with body measurement traits. Author mentioned RET gene plays an important role in gastrointestinal nervous system but there is no evidence to have direct association of this gene in cattle traits. Author found an association and mentioned two SNPs would be useful for marker assisted selection in cattle. If the SNPs were useful for marker assisted selection, I think that the SNPs have to explain a lot of genetic variation or phenotypic variation but there is no mention and analyses in this manuscript. Moreover, very small sample size was used to look at SNP association in this study. Therefore, it is very difficult to figure our effects of the SNPs to cattle body measurement.

Therefore, I recommend that author should update many things for the publication.

Specific comments

Author identified two SNPs using PCR primer (18 primers) designed based on AC_000185.1, I wonder how many SNPs in there?? And Did you find out how many SNPs in RET gene in GenBank SNP database??. It is really novel or common SNPs which are already in commercial cattle SNP chips (ex illumine 50K and 777K chip) The two SNPs are completely correlated in QC breed and highly correlated in LY, if you do SNP pruning, only one SNP will be left!! Two SNPs are same in QC breed!! Author used very simple statistical model (ANOVA) to look at association between SNPs and phenotypes. You only fit genotype effect and breed effect, is there any other fixed effect, birth year, month, there age and pedigree information?? If you have you have to fit those effect in your model In the statistical model, even if two SNPs are highly correlated, but author fit SNP separately in each analyses, I think that you should fit the two SNPs together, then see SNP effects!! You need to calculate additive and dominance effect based on this ANOVA analyses, then how much the significant SNP can explain genetic or phenotypic variation?? Please add the result to manuscript! I wondering the Table 6 is a result from haplotype association analyses??, what is combined sites ?? Author mentioned marker assisted selection in animal breeding, normally, marker assisted breeding value use additive effect of SNPs, if then, regression analysis would more proper to look at SNP effects in this study. How do you think about??

Author Response

Author identified two SNPs using PCR primer (18 primers) designed based on AC_000185.1, I wonder how many SNPs in there?? And Did you find out how many SNPs in RET gene in GenBank SNP database??.It is really novel or common SNPs which are already in commercial cattle SNP chips (ex illumine 50K and 777K chip).

Reply:We screened SNPs in all exons of RET gene in Qinchuan and Nanyang cattle by using 18 primers. Finally, we identified two SNPs that were described in the manuscript in the exon region. Although so many SNPs were found in the GenBank database, we only found two exon SNPs in these two cattle breeds. The two identified SNPs (SNP1: rs110630023, c.1407A>G and SNP2: rs109861339, c.1425C>G) were firstly used to be analyzed, therefore they were defined as novel SNPs.

The two SNPs are completely correlated in QC breed and highly correlated in LY, if you do SNP pruning, only one SNP will be left!! Two SNPs are same in QC breed!!

Reply: We are grateful for the suggestion. According to the results, the effects of these two SNPs are the same in Qinchuan breed. However, the results of linkage disequilibrium analysis shown that these two SNPs had a strong relationship. Therefore, we cannot discard any SNPs because we are not sure these two sites were working alone or together in QC breed.

Author used very simple statistical model (ANOVA) to look at association between SNPs and phenotypes. You only fit genotype effect and breed effect, is there any other fixed effect, birth year, month, there age and pedigree information?? If you have to fit those effect in your model In the statistical model, even if two SNPs are highly correlated, but author fit SNP separately in each analyses, I think that you should fit the two SNPs together, then see SNP effects!!

Reply: Thank you for your precious advice. At the period of animal sample collection, we choose all female cattle which were between 3 and 4 years old, healthy, non-pregnant female, and the different breeds were raised under the same forage and feeding management conditions and analyzed separately. Animals of each breed were selected to be unrelated for at least three generations. Therefore, this study used the reduced linear model to determine the relationship between genotypes and the various body measurement traits. Besides, we not only analyzed the single SNP but analyzed two SNPs together by combined genotypes analysis (Table 6 and 7). The results were showing the effects of two SNPs.

You need to calculate additive and dominance effect based on this ANOVA analyses, then how much the significant SNP can explain genetic or phenotypic variation??Please add the result to manuscript!

Reply: Thanks for your advice. The additive and dominance effect are important to explain genetic and phenotypic variation. However, we are sorry that these values were not be calculated because the animal samples of two breeds what we choose were unrelated for at least three generations.

I wondering the Table 6 is a result from haplotype association analyses??, what is combined sites ??

Reply: Thank you for your suggestion. We have added the haplotype analysis in the manuscript (Table 3, line 185 – line 189). In this analysis, we found that the frequency of  the haplotype (G-C) in QC cattle and the haplotype (A-G) are 0.004, which are low to be ignored. If we use the haplotype combination to do the association analyses, the haplotype combinations are Hap1/1-AACC(113), Hap1/2-AGCG(90), and Hap2/2- GGGG(12). The combined genotypes [AA-CG (4) and AG-CC (6)] were ignored. However, we think these two combined genotypes is also important for association analyses. Besides, the Hap2/3 GGCG and Hap3/3-GGCC were not existed in our data. Therefore, were choose the combined genotypes to do the association analyses. the number of each genotype were add in Table 6 and 7.

Author mentioned marker assisted selection in animal breeding, normally, marker assisted breeding value use additive effect of SNPs, if then, regression analysis would more proper to look at SNP effects in this study. How do you think about??

Reply: Thanks for your suggestion. We approve that the additive effect of SNPs should be used in marker-assisted breeding. However, it is a pity that we don’t have enough information about pedigree or accurate age. We are very sorry that the additive effect of SNPs was not used in our manuscript. Besides, we consider the key difference of the regression analysis and variance analysis is whether the independent variable is a random variable or a categorical variable. The independent variable in our manuscript is different genotype, which belongs to the categorical variable. Therefore, we believe the variance analysis is more appropriate in our study.

Reviewer 2 Report

The present paper is well written and presents a relevant topic that can be very helpful given it deals with economically important traits in local cattle. However, the description of the statistical analysis lacks the information enough for me to be able to decide whether the procedures performed are solid or not, hence, in the present form, we cannot determine about the validity and soundness of the results and the conclusions drawn from them.

Line 16-18: This sentence seems rather unconnected from the rest of the text.

Line 19: delete the word traits.

Line 22: delete the word traits.

Line 25: decapitalize the word Rearrangement.

Line 28: delete the word traits. check across the body text as it may be present more often.

The introduction is easily followable, well written and presents quite well what the paper intends to do.

Line 129: Change is to are.

Line 152: I think supplementary figures should be embedded in the body text if you pay attention to what is described in the guidelines for authors.

Line 156-158: Font is different and the caption is not centered.

Line 181: Font is different and the caption is not centered.

Material and Methods section

All the data regarding samples, DNA extraction from samples, SNP detection and genotyping seems to have been performed adequately. However, there is no information provided regarding the statistical analysis and any clue is provided about which were the methods carried out or what was done to perform the association analysis. It is true that from Tables we could infer that the least-squares mean were compared to perform the association analysis, hence a univariate general linear model or regression model could have been considered used for each of the variables considered in this study.

Apart from the lack of information regarding the test used, no information is provided regarding the testing of residuals normality assumptions; hence we do not know whether the parametric approach used in this study is correct or not.

I suggest that the authors provide more details regarding the statistical analysis and resubmit it for me to be able to determine whether the discussion and conclusion drawn after them are valid or not.

Author Response

Response to Reviewer 2 Comments

The present paper is well written and presents a relevant topic that can be very helpful given it deals with economically important traits in local cattle. However, the description of the statistical analysis lacks the information enough for me to be able to decide whether the procedures performed are solid or not, hence, in the present form, we cannot determine about the validity and soundness of the results and the conclusions drawn from them.

Line 16-18: This sentence seems rather unconnected from the rest of the text.

Reply: Thanks for your advice. This sentence mainly described the purpose of this study and explained the reason why we begin to following works.

Line 19: delete the word traits.

Line 22: delete the word traits.

Reply: The word “traits” have been deleted.

Line 25: decapitalize the word Rearrangement.

Reply:  The word has been changed in line 25.

Line 28: delete the word traits. check across the body text as it may be present more often.

Reply: The word “traits” have been deleted. We checked the body text and revised.

The introduction is easily followable, well written and presents quite well what the paper intends to do.

Reply: Thank you very much for your affirmation.

Line 129: Change is to are.

Reply: The word has been changed in line 131 - line 132.

Line 152: I think supplementary figures should be embedded in the body text if you pay attention to what is described in the guidelines for authors.

Reply: The “Supplementary” was added to line 155 and line 381.

Line 156-158: Font is different and the caption is not centered.

Line 181: Font is different and the caption is not centered.

Reply: The font has changed to match the font of text and the caption has been centered.

Material and Methods section

All the data regarding samples, DNA extraction from samples, SNP detection and genotyping seems to have been performed adequately. However, there is no information provided regarding the statistical analysis and any clue is provided about which were the methods carried out or what was done to perform the association analysis. It is true that from Tables we could infer that the least-squares mean were compared to perform the association analysis, hence a univariate general linear model or regression model could have been considered used for each of the variables considered in this study.

Apart from the lack of information regarding the test used, no information is provided regarding the testing of residuals normality assumptions; hence we do not know whether the parametric approach used in this study is correct or not.

I suggest that the authors provide more details regarding the statistical analysis and resubmit it for me to be able to determine whether the discussion and conclusion drawn after them are valid or not.

Reply: Thank you for your precious advice. The detail of statistical analysis was revised in line 128-136.

Reviewer 3 Report

The study describes the genetic variants of RET gene in 2 breeds of Chinese cattle and investigates the potential functional consequences of 2-marker-genotypes of this gene involved in body size and enteric nervous system. Study has been performed with appropriate ethical approval in place and sound experimental design.

General:

All the measured traits in both population of cattle could only be associated with the genotype but not in a causative relationship with the genetic variants. The phenotype does not present a strong difference at mean and SD level and I strongly suggest reframing from usage phrases such as  “having an effect or affecting on the traits” throughout the manuscript , instead use the significant association and keep the observation objective. The chain of causality falls outside of the remits of this study.

Specific:

line 65-67 the heroic assumption of the gene involved in the development of the gut-ganglionic network will subsequently affect the size and growth rate of the cattle is a long leap of non-evidential faith.  Is there any previous studies for support this claim if so please cite. If you have based your assumption on the human literature and Hirshsprung disease perhaps some pathology or anatomy of these animal could help your cause. I am not sure if this is possible to produce as part of this study or not.  

Line 136-140 The secondary structure prediction doesn’t show a stark difference between the wildtype and mutant version of the protein. Is there a possibility for transfection or synthesis of these 2 version of the protein and compare the neurotrophic ligand binding capabilities on the cell surface (either kidney or intestinal cell lines). Is there any other biological assay to differentiate the function of these 2 protein types as they don’t seem to be folded any differently.

Figure S1 Please explain the frequency in detail for the codon and amino acids. Is it in thousands or is it based on your own study numbers. If the frequency is based on your study could you please provide a frequency comparison statistics between the 2 genotypes of the SNP2. This statistics can further support your claim of higher usage of CCG in lines 149-150

Table 2 Under the breeds column please correct the LY to NY in both rows.

Table 2&3 &4 as the G>A please rearrange genotype similar to C>G to have GG, GA and AA instead of AA,AG and GG as its confusing and not consistence with SNP1

line 180 figure 3 is not needed as is a repetition of the D’ and R2 values mentioned in the text.

line 184 Please reframe from using “having a significant effect” as it implies causality. Association analysis should be carried out as objective as possible.

Table 6- Please separate the genotypes with “-“ for consistency with table 5

Line 237 – “and its effect on bovine body …. “ please change to significant association with.

Line 268 refer to the latter comment about affecting.

Author Response

Response to Reviewer 3 Comments

The study describes the genetic variants of RET gene in 2 breeds of Chinese cattle and investigates the potential functional consequences of 2-marker-genotypes of this gene involved in body size and enteric nervous system. Study has been performed with appropriate ethical approval in place and sound experimental design.

General:

All the measured traits in both population of cattle could only be associated with the genotype but not in a causative relationship with the genetic variants. The phenotype does not present a strong difference at mean and SD level and I strongly suggest reframing from usage phrases such as  “having an effect or affecting on the traits” throughout the manuscript , instead use the significant association and keep the observation objective. The chain of causality falls outside of the remits of this study.

 Reply:  Thank you for your comments. The phrases have been reframed in the manuscript.

Specific:

line 65-67 the heroic assumption of the gene involved in the development of the gut-ganglionic network will subsequently affect the size and growth rate of the cattle is a long leap of non-evidential faith.  Is there any previous studies for support this claim if so please cite. If you have based your assumption on the human literature and Hirshsprung disease perhaps some pathology or anatomy of these animal could help your cause. I am not sure if this is possible to produce as part of this study or not.  

 Reply:  Thanks. We are sorry that we didn’t found any previous studies to support our assumption. We mainly elaborated the relationship of the RET gene and the enteric nervous system, which affects the nutrient absorption. Therefore, we have added the reference (4) that described the relationship between ENS and nutrient absorption in line 43 - line 45. That can be more supportive of our hypothesis.

Line 136-140 The secondary structure prediction doesn’t show a stark difference between the wildtype and mutant version of the protein. Is there a possibility for transfection or synthesis of these 2 version of the protein and compare the neurotrophic ligand binding capabilities on the cell surface (either kidney or intestinal cell lines). Is there any other biological assay to differentiate the function of these 2 protein types as they don’t seem to be folded any differently.

 Reply: Thank the reviewer for these precious comments and suggestions. In our study, the result of predicted protein secondary structure and 3-D structure show little difference between the wildtype and mutant types, which may provide some references for the following research. We are regretful that we could not add more study to compare the neurotrophic ligand-binding capabilities on the cell in this manuscript because of the lack of laboratory conditions. We understand that is important, so we try to found another biological assay to compare these 2 protein types, however, it will take an extended amount of time to do the following experiment to optimize the whole experiment.

Figure S1 Please explain the frequency in detail for the codon and amino acids. Is it in thousands or is it based on your own study numbers. If the frequency is based on your study could you please provide a frequency comparison statistics between the 2 genotypes of the SNP2. This statistics can further support your claim of higher usage of CCG in lines 149-150

 Reply: Thanks for your suggestion. The usage frequency of CCG is based on the CDS region of bovine RET gene. So the frequency comparison statistics between the 2 genotypes of the SNP2 cannot be calculated.

Table 2 Under the breeds column please correct the LY to NY in both rows.

Reply: Sorry for typing. We have changed “LY” to “NY”.

Table 2&3 &4 as the G>A please rearrange genotype similar to C>G to have GG, GA and AA instead of AA,AG and GG as its confusing and not consistence with SNP1

Reply:  We are sorry for our miscalculation. We have checked the number of SNP (rs110630023 and rs109861339) in Ensemble database, the name of two SNPs have been revised (SNP1: rs110630023, c.1407A>G and SNP2: rs109861339, c.1425C>G). Considering these two SNP were located in exon 7, we choose the number of the exon to defined these SNPs. And the mutation genotype was revised by contrasting with the information of SNPs in the Ensemble database. Therefore, the rank of genotype was not rearranged in the table.

line 180 figure 3 is not needed as is a repetition of the D’ and R2 values mentioned in the text.

 Reply: Thanks, we have moved figure 3 in the supplemental materials.

line 184 Please reframe from using “having a significant effect” as it implies causality. Association analysis should be carried out as objective as possible.

 Reply: Thanks. We have reframed the phrases in line 195.

Table 6- Please separate the genotypes with “-“ for consistency with table 5

  Reply: Thanks. The “-” has been added in table 7.

Line 237 – “and its effect on bovine body …. “ please change to significant association with.

 Reply: Thanks. We have revised this sentence as follows:

“and significant association with bovine body measurement traits was researched for the first time.”

Line 268 refer to the latter comment about affecting.

Reply: Thanks. We have reframed this phrase in the manuscript.

Round 2

Reviewer 1 Report

minor comment

line 128. the statistical model is not one way ANOVA? the Model has two factors (genotype and breed).. please check it.

Line 128 - 136. your statistical model already correct breed effect in the model to look at genotype effect on body measurement. However,  table 4 and 5 described genotype effect in separate breed (QC and NY). it looks your model would be y ~ G + e (One way ANOVA).. Please check it with first comment and update

Author Response

line 128. the statistical model is not one way ANOVA? the Model has two factors (genotype and breed).. please check it.

Line 128 - 136. your statistical model already correct breed effect in the model to look at genotype effect on body measurement. However,  table 4 and 5 described genotype effect in separate breed (QC and NY). it looks your model would be y ~ G + e (One way ANOVA).. Please check it with first comment and update

Respond:Sorry for our inattention. The model is revised in line 128.

Reviewer 2 Report

Thank you for the addition of further details regarding the statistical analysis of the data. However, still no information is provided regarding the testing of residuals normality assumptions; hence we do not know whether the parametric approach used in this study is correct or not.

Please provide more information regarding assumption testing.

Author Response

Thank you for the addition of further details regarding the statistical analysis of the data. However, still no information is provided regarding the testing of residuals normality assumptions; hence we do not know whether the parametric approach used in this study is correct or not.

Please provide more information regarding assumption testing.

Respond: Sorry for our carelessness. We add the methods of the testing of residuals normality assumptions in line 135-136 and the result of the residuals normality test were shown in line 198-199 and the Supplemental Figure S3 and S4.
